# Nonparametric Covariance Regression for Massive Neural Data on Restricted Covariates via Graph

## Abstract

Modern recording techniques enable neuroscientists to simultaneously study neural activity across large populations of neurons, with capturing predictor-dependent correlations being a fundamental challenge in neuroscience. Moreover, the fact that input covariates often lie in restricted subdomains, according to experimental settings, makes inference even more challenging. To address these challenges, we propose a set of nonparametric mean-covariance regression models for high-dimensional neural activity with restricted inputs. These models reduce the dimensionality of neural responses by employing a lower-dimensional latent factor model, where both factor loadings and latent factors are predictor-dependent, to jointly model mean and covariance across covariates. The smoothness of neural activity across experimental conditions is modeled nonparametrically using two Gaussian processes (GPs), applied to both loading basis and latent factors. Additionally, to account for the covariates lying in restricted subspace, we incorporate graph information into the covariance structure. To flexibly infer the model, we use an MCMC algorithm to sample from posterior distributions. After validating and studying the properties of proposed methods by simulations, we apply them to two neural datasets (local field potential and neural spiking data) to demonstrate the usage of models for continuous and counting observations. Overall, the proposed methods provide a framework to jointly model covariate-dependent mean and covariance in high dimensional neural data, especially when the covariates lie in restricted domains. The framework is general and can be easily adapted to various applications beyond neuroscience.

## 1 Introduction

Modern neural recording techniques, such as high-density silicon probes (Jun et al., 2017; Steinmetz et al., 2021; Marshall et al., 2022) and large-scale calcium imaging methods (Ahrens et al., 2013; Kim et al., 2016; Grienberger et al., 2022), allow us to obtain massive neural activity data across different regions. Capturing heteroscedasticity in multivariate processes and correlations among these neurons, potentially along with experimental stimulus (e.g., visual gratings) or animal behaviors (e.g., animal locations and movement speed) as covariates, is important for providing scientific insight.

Given the prevalence of time series data in neuroscience, numerous methods have been developed for time series modeling. For example, to capture correlation patterns of high-dimensional neural data, several unsupervised latent factor models have been widely used in neuroscience community, either 1) assuming the latent factors evolve linearly with a Gaussian noise (Macke et al., 2011) or 2) modeling the progression of latent factors more generally by a Gaussian process (Yu et al., 2009). These two modeling strategies leads to two fundamental latent factor models: 1) linear dynamic systems model (LDS, i.e., the dynamic latent factor model) and 2) Gaussian process factor analysis model (GPFA). Based on these models, we can further include the covariates into the model (may also model smoothness of parameters by LDS/ GP), to study the relationship between neural activity and interested features. However, these methods only model the dynamics of mean and assume homoscedastic across time and covariates. Although it's possible to reduce residual correlation over time/ covariates by more intricate modeling, additional complexity is not efficient and may lead to

overfitting of the data. Moreover, experimental findings suggest that the variability of neural activity changes over time and across experiment settings (Churchland et al., 2010; 2011), potentially providing information about the external world beyond the average neural activity, such as signature of decision making (Churchland et al., 2011), movement preparation (Churchland et al., 2006), or stimulus onset (Churchland et al., 2010).

Therefore, our focus here is on mean-covariance modeling for high dimensional neural data, considering either continuous or categorical experiment covariates. In the context of time series analysis, which is a special case, modeling volatility (conditional standard deviation) has a long history (Chib et al., 2009), including multivariate (generalized) autoregressive conditional heteroscedasticity (GARCH, Engle (1982); Bollerslev (1986); Engle (2002)), multivariate stochastic volatility models (Harvey et al., 1994) and Wishart process (Philipov and Glickman, 2006b;a; Gourieroux et al., 2009). Within computational neuroscience community, some LDS-based methods such as dynamic Conway-Maxwell model (Wei and Stevenson, 2023), allow for over- and under-dispersion by dynamic modeling of both mean and dispersion parameter on covariates for neural spikes (counting time series). These models can potentially handle high-dimensional neural recordings by incorporating dynamic latent factors, although the parametrization may not be efficient, and the Markov assumption may be inappropriate. In general context, Nejatbakhsh et al. (2023) recently proposed a model based on (Gaussian-)Wishart process (Philipov and Glickman, 2006b; Gourieroux et al., 2009) especially for repeated trials, to handle the smoothness of mean and covariance over covariates. However, their method may have poor performance in the case with massive neurons. Beyond the neuroscience community, the research on large-scale mean-variance regression also has a long history. Some classical strategies rely on regression to elements of log or Cholesky decomposition of conditional covariance (Chiu et al., 1996; Pourahmadi, 1999; Leng et al., 2010; Zhang and Leng, 2012), which are ill-suited for high dimensional data. Instead, Hoff and Niu (2012) modeled covariance as a quadratic function on covariates with baseline, though parametric assumption may limit the usage of the model. Fox and Dunson (2015) proposed a Bayesian non-parametric model for continuous response, by assuming both latent factor and basis of factor loading are covariate-dependent, and handle the smoothness by GP. Some methods, such as Franks and Hoff (2019), Wang et al. (2019) and Franks (2022) have also been proposed for high-dimensional covariate settings ($p \gg N$).

However, there are still several challenges, especially for neuroscience applications. First, the spike count data are majorly used for studying neural activity, but the counting observations make the inference intractable. Second, many covariates, such as animal locations and movement orientations, fall in restricted subspace, ignoring the subspace information may lead to inappropriate mean-covariance inference. To address these problems, motivated by Fox and Dunson (2015) and Dunson et al. (2022), we introduce a latent factor covariance regression model, based on graph properties of covariates, and the model is flexibly inferred by an efficient MCMC algorithm.

The rest of this paper is structured as follows. In section 2, we introduce the mean-covariance regression model for high dimensional neural data with restricted inputs, considering both continuous and counting observations. The key steps of MCMC algorithm for sampling posterior distributions are provided. Then, after validating and studying the proposed methods using synthetic datasets in Section 3, we apply our methods to two neural data (local field potential and neural spikes data) in Section 4, to illustrate usage with continuous and counting observations. Finally, in section 5, we conclude with some final remarks and discuss some potential extensions of our current model for future research.

## 2 Method

In this section, we first introduce the covariance regression models in latent space for high dimensional neural data. Both continuous and counting response are considered. We then introduce the graph based Gaussian process to account for restricted inputs, which are commonly encountered in many applications. The models are inferred by an MCMC algorithm, and we briefly outline several key steps. See Section A and B for more details on prior specification and MCMC steps for model inference. The code for MATLAB implementation can be found in the supplementary material.

## 2.1 Covariance Regression Model for High Dimensional Neural Data

Denote the neural activity of $n$ neurons in experiment condition $j$ as $\boldsymbol{y}_j = (y_{1j}, \ldots, y_{nj})'$, for $j = 1, \ldots, p$. For continuous response, we model it by a multivariate Gaussian nonparametric mean-covariance regression model as

$$\boldsymbol{y}_j = \boldsymbol{\mu}(\boldsymbol{x}_j) + \boldsymbol{\epsilon}_j$$

, where $\boldsymbol{x}_j \in \mathbb{R}^q$ is the covariate, $\boldsymbol{\epsilon}_j \sim N_n(0, \Sigma(\boldsymbol{x}_j))$ and $\boldsymbol{\epsilon}_j$s are independent. Then the mean and covariance are $\mathrm{E}(\boldsymbol{Y}_j) = \boldsymbol{\mu}(\boldsymbol{x}_j)$ and $\mathrm{Cov}(\boldsymbol{Y}_j) = \Sigma(\boldsymbol{x}_j)$. For counting data such as neural spikes, we model it in a Poisson GLM with log-link as

$$\boldsymbol{y}_j \mid \boldsymbol{\lambda}_j \sim \mathrm{Poisson}(\boldsymbol{\lambda}_j)$$
$$\log \boldsymbol{\lambda}_j = \boldsymbol{\mu}(\boldsymbol{x}_j) + \boldsymbol{\epsilon}_j$$

Then, the mean and covariance of response in the Poisson log-normal model (Aitchison and Ho, 1989), which allows for modeling over-dispersion, are

$$\mathrm{E}(\boldsymbol{Y}_j) = \boldsymbol{\Lambda}_j = \exp\left[\boldsymbol{\mu}(\boldsymbol{x}_j) + \frac{1}{2}\mathrm{diag}\left(\Sigma(\boldsymbol{x}_j)\right)\right]$$

$$\mathrm{Cov}(\boldsymbol{Y}_j) = \boldsymbol{\Lambda}_j + \boldsymbol{\Lambda}_j \left[\exp \Sigma(\boldsymbol{x}_j) - \mathbf{1}\mathbf{1}'\right] \boldsymbol{\Lambda}_j$$

To save notations, we denote the (pseudo) response, either transformed or not, as $\boldsymbol{\zeta}_j$ such that $\boldsymbol{\zeta}_j = \boldsymbol{y}_j$ for Gaussian case and $\boldsymbol{\zeta}_j = \log \boldsymbol{\lambda}_j$ for Poisson case.

To handle large number of neurons $n$, which are common for modern neural recording techniques, we resort to model proposed by (Fox and Dunson, 2015) so that the response is induced through a factor model as

$$\boldsymbol{\zeta}_j = \Lambda(\boldsymbol{x}_j)\boldsymbol{\eta}_j + \boldsymbol{\epsilon}_j$$
$$\boldsymbol{\eta}_j \sim N_k(\psi(\boldsymbol{x}_j), I_k)$$
$$\boldsymbol{\epsilon}_j \sim N_N(\mathbf{0}, \Sigma_0)$$

, where $\Lambda(\boldsymbol{x}) \in \mathbb{R}^{n \times k}$ for $k \ll n$ and $\Sigma_0 = \mathrm{diag}(\sigma_1^2, \ldots, \sigma_n^2)$. Then by marginalizing out $\boldsymbol{\eta}_j$, the mean and covariance for the model are

$$\boldsymbol{\mu}(\boldsymbol{x}_j) = \Lambda(\boldsymbol{x}_j)\psi(\boldsymbol{x}_j)$$
$$\Sigma(\boldsymbol{x}_j) = \Lambda(\boldsymbol{x}_j)\Lambda(\boldsymbol{x}_j)' + \Sigma_0$$

If we use time as covariates, and model the progression of $\boldsymbol{\eta}_j$ either by linear dynamics or Gaussian process, the model reduces to dynamic factor analysis model/ linear dynamic systems (LDS, Macke et al. (2011)) or Gaussian process factor analysis (GPFA, Yu et al. (2009)) model that is widely used for time series data, especially in neuroscience.

Estimating high dimensional factor loading $\{\Lambda(\boldsymbol{x}_j)\}$ can be difficult ($nkp$ parameters), therefore we further factorize the loading as in Fox and Dunson (2015) to reduce the dimension, such that

$$\Lambda(\boldsymbol{x}_j) = \boldsymbol{\Theta}\boldsymbol{\xi}(\boldsymbol{x}_j)$$

, where $\boldsymbol{\Theta} \in \mathbb{R}^{n \times L}$ and $\boldsymbol{\xi}(\boldsymbol{x}_j) \in \mathbb{R}^{L \times k}$, for $L \ll n$. In this paper, we fix the factor dimension $k$, but allow the basis size $L \to \infty$ and put multiplicative shrinkage prior (Bhattacharya and Dunson, 2011) to adaptively choose $L$.

To capture the smoothness of response mean and covariance among different experiment conditions $j$, we can put Gaussian process (GP) priors on both latent factor and loading basis. Specifically, for $\psi(\boldsymbol{x}) = (\psi_1(\boldsymbol{x}), \ldots, \psi_k(\boldsymbol{x}))'$, we have $\psi_m \sim \mathrm{GP}(0, c_\psi)$ independently for $m = 1, \ldots, k$. For factor loading basis, let $\xi_{lm}(\boldsymbol{x})$ be element of $\xi(\boldsymbol{x})$, and we put GP priors on each element independently with the same kernel such that $\xi_{lm} \sim \mathrm{GP}(0, c_\xi)$. This leads to a Wishart process in a manner slightly different from that described in (Nejatbakhsh et al., 2023). The covariances for $\psi_m(\boldsymbol{x})$ and $\xi_{lm}(\boldsymbol{x})$ are both unit, i.e. the correlation, for model identifiability ( Cai et al. (2023); Conti et al. (2014), although this is not necessary in this paper since we focus on estimation of mean and covariance). To further take the restricted input into account, we construct the covariance by incorporating intrinsic geometry of subspace. This leads to Graph Laplacian-based GP (GL-GP) and we discuss details in the next subsection. Moreover, by using GP or GL-GP priors, we can easily impute/ sample the missing response under certain conditions, based on conditional Gaussian distribution. See more details on model settings and prior specification in appendix A.

## 2.2 Graph Based GP for Restricted Inputs

In neuroscience experiment and many other applications, it is common for the inputs to fall in a restricted space. For example, in target reaching experiments, both the targets and animals movements are confined to small areas Wise (1985); Galiñanes et al. (2018), and in maze running experiment, the animals are even restricted to move along the pre-designed path (Mizuseki et al., 2013). These restrictions are usually not easy to transform into a non-restricted space, and in some situation the restriction cannot be known in advance, e.g. because of animals internal preferences.

Ignoring these restrictions may lead to a sub-optimal results. For instance, two points that are close in Euclidean space might be far apart in a restricted space, and modeling in the Euclidean space may lead to inappropriate smoothness. Several methods have been proposed to address these issues within the framework of GP regression. When the restricted space is a known submanifold, we can extrinsically embed the manifold in a higher-dimensional Euclidean space (Lin et al., 2019), or intrinsically approximate heat kernel (Niu et al., 2019) by Monte Carlo or use other valid kernels (Li et al., 2023; Borovitskiy et al., 2020). For an unknown submanifold, we can instead employ locally linear regression methods (Cheng and Wu, 2013), and we can also use similar ideas in semi-supervised approaches (Zhu et al., 2003; Zhu, 2005; Belkin et al., 2006; Nadler et al., 2009; Dunlop et al., 2020; Wang and Lerman, 2015). However, whether known or unknown, these methods assume the subspace is a manifold, which may not be appropriate in some cases.

Here, we use the kernel based on Graph Laplacian (GL) proposed by Dunson et al. (2022), and hence both $\psi(\boldsymbol{x})$ and $\xi(\boldsymbol{x})$ are modeled by GL-GPs, denoted $\psi_m \sim \mathrm{GLGP}(0, c_\psi)$ and $\xi_{lm} \sim \mathrm{GLGP}(0, c_\xi)$. The GL-GP incorporates intrinsic geometry of restricted space (not necessarily a manifold) by taking finitely many eigenpairs of the GL, whose covariance approximates a diffusion process on restricted space based on intrinsic distances between data points. Specifically, given a kernel $c(\boldsymbol{x}, \boldsymbol{x}')$ (simply use the squared exponential kernel $c(\boldsymbol{x}, \boldsymbol{x}') = \exp\left(-||\boldsymbol{x} - \boldsymbol{x}'||_2^2 / 4\kappa\right)$ in this paper), the GL matrix is defined as $L_G = (D^{-1}W - I)/\kappa$. Here, $W = \{W_{ij}\} \in \mathbb{R}^{p \times p}$ is an affinity matrix defined by the kernel as $W_{ij} = c(\boldsymbol{x}, \boldsymbol{x}')/(r(\boldsymbol{x}_i)r(\boldsymbol{x}_j))$ with $r(\boldsymbol{x}) = \sum_{i=1}^{p} c(\boldsymbol{x} - \boldsymbol{x}')$, and $D \in \mathbb{R}^{p \times p}$ is a diagonal matrix with $i$th diagonal entry be $D_{ii} = \sum_{j=1}^{p} W_{ij}$, and $\kappa$ is the same as used in kernel $c(\boldsymbol{x}, \boldsymbol{x}')$. Using the constructed $L_G$, we can define the covariance matrix as

$$\tilde{H} = p \sum_{i=1}^{K-1} e^{-\mu_{i,p,\epsilon}} \tilde{\nu}_{i,p,\epsilon} \tilde{\nu}_{i,p,\epsilon}^{\mathsf{T}}$$

, where $\mu_{i,p,\epsilon}$ is eigenvalue of $-L_G$, $\tilde{\nu}_{i,p,\epsilon}$ is corresponding eigenvector and $\{\epsilon, K, t\}$ are tuning parameters (estimated by MLE or sampled by MCMC, see details in Section 2.3). We further convert the covariance matrix to correlation as $H$, for model identifiability. For more details of GL-GP, including the effects of three tuning parameters ($\{\epsilon, K, t\}$), see Dunson et al. (2022).

## 2.3 Inference

The model is inferred by a MCMC algorithm. The sampling details can be found in the appendix Section B, and we provide some key sampling strategies here.

First, the sampling of the Poisson model for count data (e.g. neural spikes) can be intractable because of non-conjugacy. However, we can approximate a Poisson distribution by a negative binomial (NB) distribution, using the fact that $\lim_{r \to \infty} \mathrm{NB}\left(r, \sigma(\zeta - \log r)\right) = \mathrm{Poisson}(e^\zeta)$, where $\sigma(\zeta) = e^\zeta/(1 + e^\zeta)$ and $\mathrm{NB}(r, p)$ denotes the NB distribution with expectation be $rp/(1 - p)$. By using a large enough dispersion parameter $r$, we can treat a Poisson distribution as an NB distribution, which follows the Pólya-Gamma (PG) augmentation scheme (Windle et al., 2013; Polson et al., 2013). Specifically, for response from neuron $i$ under condition $j$, we introduce an auxiliary variable $\omega_{ij} \sim \mathrm{PG}(r_{ij} + y_{ij}, \zeta_{ij} - \log r_{ij})$, where $\mathrm{PG}(a, b)$ denotes the Pólya-Gamma distribution. Then we can sample the pseudo response $\zeta_{ij} \sim N(m_{ij}, V_{ij})$, where $V_{ij} = \left(\omega_{ij} + \sigma_i^{-2}\right)^{-1}$, $m_{ij} = V_{ij}(\kappa_{ij} + \sigma_i^{-2}\mu_{ij})$ and $\kappa_{ij} = (y_{ij} - r_{ij})/2 + \omega_{ij} \log(r_{ij})$. With samples of $\zeta_{ij}$, the sampling for other parameters is the same as in the Gaussian case.

Second, even by factorizing the loading as $\Lambda(\boldsymbol{x}) = \boldsymbol{\Theta}\boldsymbol{\xi}(\boldsymbol{x})$, sampling $L \times k \times p$ parameters for $\boldsymbol{\xi}(\boldsymbol{x})$ directly from the joint posterior can be infeasible when $p$ is large (i.e., many experimental conditions). This is especially common for neural data analysis, since the recording length can be

very long. In other words, if we take the timestamp as a covariate and try to track the mean and covariance along the time (and potentially experimental settings), the sampling procedure can be very cumbersome. Therefore, we sample each element of $\boldsymbol{\xi}(\boldsymbol{x})$ sequentially. This procedure can be looped multiple times within each MCMC iteration to achieve better mixing.

Third, although Fox and Dunson (2015) claims that the model is relatively robust to the GP hyper-parameters because of quadratic mixing over GP dictionary elements, we observe that the inference can be sensitive to hyper-parameters ($\{\epsilon, K, t\}$) for GL-GP. Here, for modeling latent factor and loading with GL-GP prior, we can fix the hyper-parameters by maximizing likelihood, marginally or conditionally on $\psi(\boldsymbol{x})$ and $\zeta(\boldsymbol{x})$ according to burn-in samples, or sample them in each iteration (again marginally or conditionally on samples of $\psi(\boldsymbol{x})$ and $\zeta(\boldsymbol{x})$). Here, we found that sampling on marginal distribution is computationally cumbersome, and hence using slice sampler may not be feasible for large datasets (Murray and Adams, 2010). To choose the hyper-parameters more efficiently, we can also use a data-driven heuristic, based on the fact that the autocorrelation $\mathrm{ACF}(x) = \mathrm{corr}(\Sigma_{ij}(0), \Sigma_{ij}(x))$ is specified by the kernel function (Fox and Dunson, 2015).

# 3 SIMULATIONS

All the following experiments, including applications in Section 4, are conducted on a laptop with an Intel(R) Core(TM) i7-8665U CPU @ 1.90GHz 2.11 GHz. Here, we simulate recordings of 50 neurons, for both continuous and counting responses, when an animal is restricted to move within a "two boxes linking with a tunnel" area, i.e., two squares with side length 3, connected with a 2-by-1 rectangle. The input features $\boldsymbol{x} \in \mathbb{R}^2$ are coordinates within the restricted domain. The response is generated from the model in Section 2.1 for both Poisson and Gaussian cases, with $L = 4$ and $k = 2$. The loading coefficients $\Theta$ are independently generated by a Gaussian distribution. The latent factor mean ($\psi_m(\boldsymbol{x})$) and loading basis ($\zeta_{lm}(\boldsymbol{x})$) are generated by mixtures of scaled Gaussian density. Here, we can observe response from 100 random locations, and our goal is to infer mean and covariance for the whole restricted area, including 1000 other random locations as test (held-out). We independently generate the data and conduct analysis six times to check robustness of our methods (Section D.1), and one set of data is illustrated in Figure 1A and D.

For each set of simulated training data, we fit four models: Gaussian process Wishart process (GPWP) model (Nejatbakhsh et al., 2023) (on a single trial), latent GP model (L-GP), latent GL-GP model with fixed hyper-parameters (L-GLGP-fixed) and latent GL-GP model with hyper-parameters sampled (L-GLGP-adaptive). All kernels used here are squared exponential for fair comparisons. The tuning parameters for GPWP are selected by 5-fold cross-validation. The bandwidths for L-GP are sampled within MCMC, and the samples from L-GP are used to select the hyper-parameters for L-GLGP-fixed. For all latent factor models (L-models), we use both 1) $L = 10, k = 2$ and 2) $L = 10, k = 5$. The running time can be found in the Appendix C. For all six experiments, the held-out log-likelihood for latent factor models, using either $k = 2$ or $k = 5$ is shown in Section D.1, and one set of them is visualized in Figure 1B and E. For this set of experiment, the fitted mean and covariance latent factor based models (L-GP, L-GLGP-fixed and L-GLGP-adaptive) are projected to the first three principal components of the ground true mean response, which captures more than 90% variance of the data. The fitting results corresponding to Figure 1B and E in PC space for $k = 2$ are shown in Figure 1C and A1, and for $k = 5$ in Figure A2 (Gaussian) and A3 (Poisson).

In all cases, the latent factor based models perform better than GPWP models in terms of held-out likelihood. Moreover, the latent factor models are easier to fit, as tuning hyper-parameters for GPWP via cross-validation is cumbersome and difficult, especially for Poisson response. We found that the last few columns of fitted loading basis $\Theta$ are shrunk to 0 for latent factor based models, suggesting $L = 10$ is large enough. In the Gaussian case, the latent factor based models are robust to latent dimension $k$, which is consistent with previous findings (Fox and Dunson, 2015). The L-GLGP models (fixing or sampling hyper-parameters) generally improve the fitting results slightly compared to L-GP, either quantitatively for held-out log-likelihood or qualitatively by visualizing fitted mean and covariance. However, for the Poisson case, the L-GLGP-adaptive models are more robust to $k$. Specifically, the held-out log-likelihood for $k = 2$ and $k = 5$ for L-GLGP-adaptive are closer (Section D.1 and Figure 1). The inferred mean and covariance for L-GP with $k = 5$ can be

noisy, while those for L-GLGP-adaptive are relatively close to the ground truth for both $k = 2$ and $k = 5$.

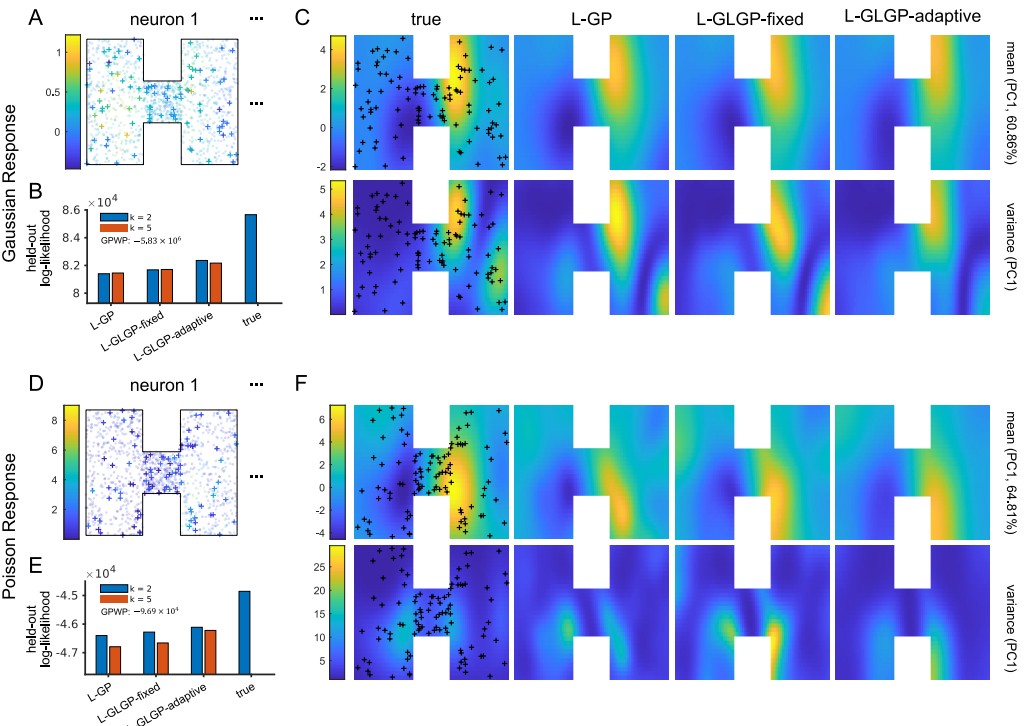

Figure 1: **Simulations.** Here, we simulate Gaussian and Poisson response in a "boxes connecting with a tunnel" restricted area, and fit response from 100 observed locations to the proposed model. The same procedures are replicated five time, with one set of results for Gaussian response are summarized in A-C and Figure A1A and A2, while the results for Poisson response can be found in D-E and Figure A1B and A3. The results in terms of held-out log-likelihood for all experiments are summarized in Section D.1. (**A**) and (**D**) The response from observed (100 plus signs) and held-out (1000 circle signs) points, for neuron 1 as examples. (**B**) and (**E**) The log-likelihood for held-out dataset (constant drop), fitting with GPWP,L-GP, L-GLGP with fixed hyper-parameters (L-GLGP-fixed) and L-GLGP with hyper-parameters sampled (L-GLGP-adaptive), with $L = 10, k = 2$ and $L = 10, k = 5$. (**C**) and (**F**) The true and fitted mean and covariance in the first PC space, with $L = 10$ and $k = 2$. The observed locations are overlaid, and the variances explained by PCs are shown alongside. The results projected in the second and third PC space are shown in Figure A1.

## 4 APPLICATIONS

We then apply the proposed methods to two neural datasets, i.e., 1) the local field potential data across the mouse brain during a visual behavior task (LFP dataset, Steinmetz et al. (2019)), to show an example of Gaussian response and 2) the multi-shank silicon probe recordings from hippocampus of a rat running back-and-forth along a linear maze (HC dataset, Mizuseki et al. (2013)), to show an example of Poisson case. In all following cases, we use squared exponential kernel and the inputs are standardized for latent factor models (L-GP and L-GLGP-fixed/adaptive). The $k$ in the latent factor models are chosen by 5-fold cross-validation for L-GP on short chains, and $L$ is large enough to ensure the last few columns of $\Theta$ are 0s.

### 4.1 LFP DATASET

In the LFP dataset, the neural activity across multiple brain regions is recorded when the mice perform a task on choosing the side with the highest contrast for visual gratings. The data contains

39 sessions from 10 mice, and each session contains multiple trials. Time bins for all measurements are 10 ms, starting 500 ms before stimulus onset. The recording ends at 2000 ms after the stimulus, and hence each trial contains data from 250 time points. See Steinmetz et al. (2019) for more details of the LFP dataset.

To show the application of proposed methods with Gaussian response, we study the relationship between LFP response and pupil conditions (area and location). This is motivated by some previous research in monkey, which found that the pupil size of monkeys can reflect neural activity (Joshi and Gold, 2020), containing LFPs, in several brain regions, including 1) cortical modulation of the pretectal olivary nucleus (PON) (Gamlin et al., 1995), 2) the superior colliculus (SC) (Wang et al., 2012; Krauzlis et al., 2013; McDougal and Gamlin, 2015) and 3) the locus coeruleus (LC) - norepinephrine (NE) neuromodulary system (Alnæs et al., 2014; Joshi et al., 2016; Wang et al., 2012; Liu et al., 2017). Moreover, the eye position of monkey are related to neural activity in superior colliculus (SC), although the position tuning is more common with build-up or burst activity and less common in neurons with visual activity (Campos et al., 2006).

Here, we choose LFP recordings from the 13th session, which include LFPs from 14 areas. These regions contain 1) the midbrain reticular nucleus (MRN), sensory and motor superior colliculus (SCs, Scm) in the midbrain, 2) the secondary motor area (MOs) in the cerebral cortex, and 3) the zona incerta (ZI) in the hypothalamus. All these areas have been found to have significant inputs to LC-NE neurons (Breton-Provencher et al., 2021). Here, we use 4 trials (trials 7-10), and 70% of these 1000 data points are used as training while remaining are held out as the test dataset (Figure 2A). In other words, the dimension of training response is $14 \times 700$ ($n = 14, p = 700$) and testing is $14 \times 300$. In the training, each iteration takes 3.5 seconds. Three pupil covariates (area, horizontal and vertical position, i.e. $q = 3$, Figure 2B and C) are considered in the model, and they are standardized before model fitting. Three models are used (L-GP, L-GLGP-fixed/adaptive), where $k = 4$ and $L = 5$. In this case, the pupil locations and areas are quite restricted, the L-GLGP performs better than L-GP, and the model is further improves by sampling hyperparameters in MCMC, according to the held-out log-likelihood (Figure 2D).

We then check the mean and variance patterns obtained from the L-GLGP-adaptive for pupil locations in the first 2 PC spaces, under three different pupil areas (0.02, 0.05 and 0.08). The evaluated location boundaries are determined by the 0.05 and 0.95 quantiles of data (Figure 2E and F). According to the fitting results, these neurons may tend to be more focused on center locations when the pupil area is relatively small (Figure 2E). The variance of LFP is larger when the pupil is smaller (area = 0.02) or larger (area = 0.08) than in the common case, but this may be caused by a lack of data for extreme pupil diameters (Figure 2F). For a more concrete conclusion for formal analysis, we may need to include more data.

## 4.2 HC Dataset

In the HC dataset, CRCNS hc-3 (Mizuseki et al., 2013), a rat was running back and forth along a 250 cm linear track. Extracellular spiking activity was recorded in the dorsal hippocampus using multi-shank silicon probes. Spikes were sorted using KlustaKwik followed by manual adjustment (Rossant et al., 2016). Here, we use data from one recording session (ec014-468) and analyze spike counts in 200 ms bins. For further details on how the data were obtained, see Mizuseki et al. (2014). In this analysis, the recordings from 14 min to 16 min are used, which contains around 4 cycles. The neurons with firing rate less than 1 Hz are discarded, which results in 36 neurons remaining.

We randomly choose 80% of points to be training and the remaining data to be test, and hence the dimension of training response is $36 \times 480$ ($n = 36, p = 480$) and testing is $36 \times 120$. Both time and (vertical) position are used as covariates ($q = 2$). Each iteration takes 3.3 seconds during training. Many neurons in the hippocampus are both position and direction tuned, and the place field of neurons may vary over time. To accommodate the effects of position and direction, we use the circular representation. In Figure 3A, we show both the original and directionally represented maze trace, with spiking counts from 4 neurons overlaid. The overall neural spikes from these 36 neurons are shown in Figure 3B. To track the dynamics of mean and variance of neural spikes, Wei and Stevenson (2023) previously proposed a dynamic Conway-Maxwell Poisson model (dCMP), which allows for both over- and under-dispersion over time. However, their method can be difficult for multi-neuron modeling in this case. Here, every neuron has the same input (time and directional

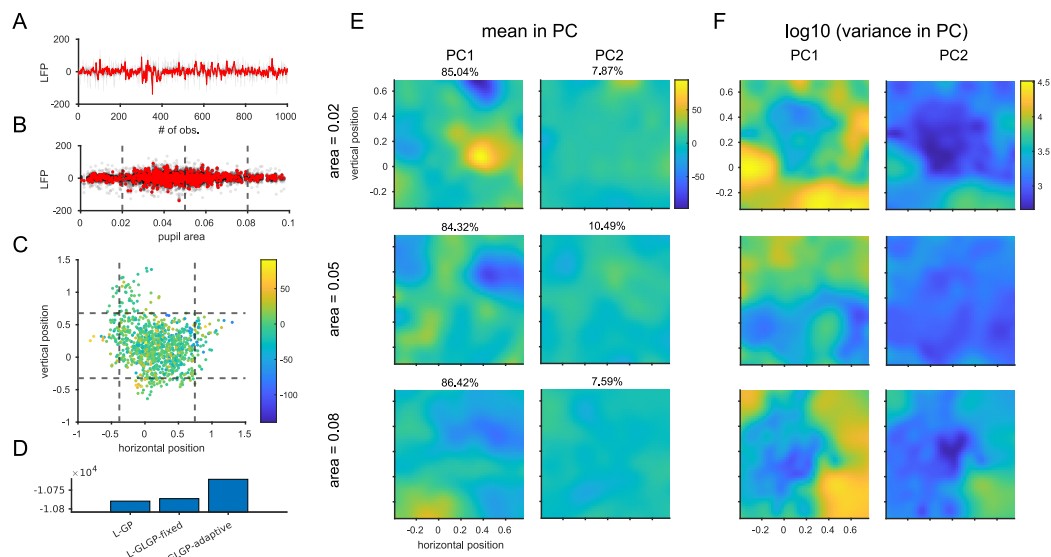

Figure 2: **Application to Steinmetz Data.** We use LFP data in 14 brain regions, from 4 trials. Each trial contains data from 250 time points. **A**. The LFP from 14 regions for all 1000 time points. The red line show SCs as an example. **B**. The scatter plot of LFP against pupil area, taking LFP from SCs (red) as an example. **C**. The pupil positions of data, colored by amplitude of LFP response. **D**. We fit 3 models (L-GP, L-GLGP-fixed/adaptive) to 70% of data, and compare the held-out log-likelihoods. The fitted mean **E** and variance **F** in PC space by L-GLGP-adaptive, according to pupil locations and 3 areas. The variance explained by PCs are shown alongside.

position); therefore under their model framework, we either 1) assume all neurons have the same response, which is inappropriate; or 2) fit the model separately for each neuron, which ignores the correlation between neurons. Here, we fit 1) dCMP separately for each neuron, 2) L-GP, 3) L-GLGP-fixed and 4) L-GLGP-adaptive. For all latent factor models, we use $k = 8$ and $L = 7$. We compare the model according to log-likelihood on test dataset, since the Poisson is nested within CMP distribution. The held-out log-likelihood for independent dCMP is $-9.90 \times 10^3$, and they are $-6.24 \times 10^3$ (L-GP), $-6.24 \times 10^3$ (L-GLGP-fixed), and $-5.89 \times 10^3$ (L-GLGP-adaptive) for the remaining models.

The fitted mean and variance in the first four PC spaces of mean is shown in Figure 3C and D. The mean response patterns in PC space correspond to 4 typical neuron in hippocampus, shown in Figure 3A (PC x corresponds to neuron x). Specifically, these are neurons that fire 1) frequently without preference of location and direction (interneurons, PC1 - neuron1), 2) selectively at 150 cm upward (PC2 - neuron2), 3) selectively downward in early cycle (PC3 - neuron3) and 4) selectively at 150cm downward (PC4 - neuron4). The corresponding variances for downward direction are generally smaller than upward ones, but the variance pattern are relatively static for these 4 cycles. Generally, both mean and variance of neural responses are tuned according to location and direction, and the patterns (especially mean) for some neurons drift along time, even in such a short period.

## 5 DISCUSSION

In this paper, we introduce a covariance regression model for high dimensional neural data, accounting for both continuous and counting observations. To accommodate the restricted experimental inputs/ covariates, we consider using a graph based Gaussian process (GLGP), to model the smoothness over covariates for both loading basis and latent factors. The model is inferred by an MCMC algorithm, where the counting observations are handled by a Pólya-Gamma (PG) data augmentation technique. After validating and studying the proposed methods by simulations, we apply them to two publicly available datasets to illustrate the usage of models with both continuous (LFP dataset) and counting observations (HC dataset).

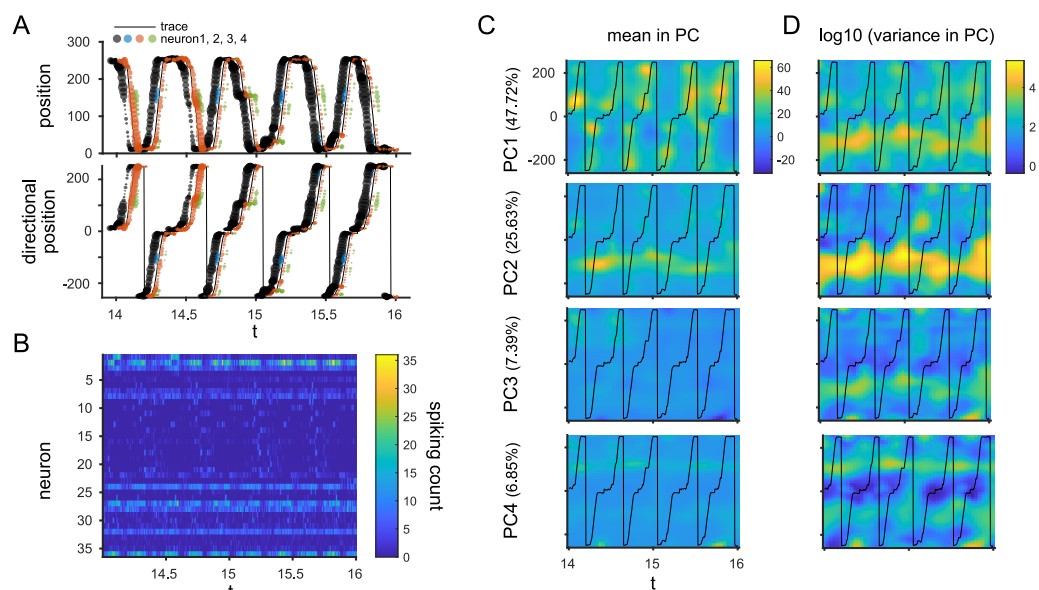

Figure 3: **Application to Hippocampus Data. A**. The trace of linear track, which contains around 4 cycles. The colored dots represent spiking activity from 4 neurons, with dot sizes proportional to spiking counts. To encode the direction into model, we use the directional representation of position here (lower panel). **B**. The spiking counts for all neurons in the data. The fitted mean and variance according to time and location in PC space are shown in **C** and **D**, with the trace of linear track overlaid. The variance explained by PC are shown alongside.

Although the proposed method can successfully model the mean and covariance according to covariates, there are some potential improvements. First, we assume independent GPs on covariates for each basis and factor dimension, to achieve computational efficiency. However, this assumption may miss some important covariance structures for different objects/ neurons, and similar results are found in gene expression data (Cai et al., 2023). Therefore, it would be attractive to model using multi-output Gaussian process (MOGP) whenever computationally feasible, either simply by linear combinations of independent GP, such as linear model of coregionalization (Philipov and Glickman, 2006b) or convolution model (Alvarez and Lawrence, 2008), or using more advanced spectral mixture to handle cross-covariance (Ulrich et al., 2015; Parra and Tobar, 2017). Second, even under independent GP assumption, the MCMC sampling can be cumbersome for large scale dataset, which is common in neural data analysis (e.g. long recordings of spiking data). The main reasons for using sampling method are checking exact posterior distributions and choosing basis dimension $L$ flexibly. However, in applications to massive data, approximation by variational inference, using methods to reduce computational cost of GP covariance matrix inversion (e.g., Zhu et al. (2024)) or using special cases of GP whenever appropriate (e.g. use linear dynamics for time series data) can be useful. Third, even though the basis dimension $L$ is chosen adaptively by shrinkage prior, we still need to specify latent dimension $k$ in advance, which may influence the Poisson model significantly (at least for L-GP). Instead, we can further sample the number of latent factors by birth-death MCMC (BDMCMC, Stephens (2000)), as in Fokoué and Titterington (2003), which requires very little mathematical sophistication and is easy for interpretation. Besides BDMCMC and shrinkage prior used for $L$, there are several other ways for choosing latent dimension, such as using multiplicative exponential process prior (Wang et al., 2016), Beta process prior (Paisley and Carin, 2009; Chen et al., 2010) or Indian Buffet process prior(Knowles and Ghahramani, 2007; 2011; Ročková and George, 2016) on the loading matrix in the Gaussian factor analysis model. However, these methods can be difficult in our case, since we further factorize the loading with basis. Finally, we observe that in Poisson version of the model, the L-GLGP can sometimes be sensitive to hyper-parameters for covariance functions. To stabilize the L-GLGP, we can assume several constraints/ priors based on initial fitting of L-GP, or use a slice sampler for these hyper-parameters (Murray and Adams, 2010) to achieve better mixing, if the computation is feasible.

Overall, as the scale of data becomes large (e.g., simultaneously observing many neurons), it can be challenging to estimate the mean and covariance (either across subjects/neurons or across covariates). Moreover, the constraints on input space/covariates make the inference more difficult. Therefore in this paper, we build a framework to accommodate both problems for continuous and counting observations. The proposed methods are quite general, and they have potential for application to data beyond neuroscience.

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

## A    PRIOR SPECIFICATION

In this section, we provide prior specifications for model parameters.

1. idiosyncratic noise $\sigma_i$: $\sigma_i^{-2} \sim \text{Gamma}(a_\sigma, b_\sigma)$

2. loading basis $\Theta$: to adaptively choose loading basis size, we use the shrinkage prior (cite) for $\Theta$ as $\theta_{il} \sim N(0, \phi_{il}^{-1}\tau_l^{-1})$, where $\phi_{il} \sim \text{Gamma}(\gamma/2, \gamma/2)$, $\tau_l = \prod_{h=1}^{l} \delta_h$, $\delta_1 \sim \text{Gamma}(a_1, 1)$ and $\delta_h \text{Gamma})(a_2, 1)$ with $h \geq 2$, $a_2 > 1$.

3. factor loading mean $\psi_m$ and loading basis $\xi_{lm}$. We assume independent (GL)-GP prior for each dimension of $\psi_m(\boldsymbol{x}_1), \ldots, \psi_m(\boldsymbol{x}_1) \sim N(\boldsymbol{0}, \boldsymbol{K})$, where the covariance is determined by GP kernel, and graph Laplacian of input $\{\boldsymbol{x}_j\}$. We also use (GL)-GP prior for $\xi_{lm}(\boldsymbol{x}_j)$, but with different hyper-parameters (even potentially with different kernels). Throughout this paper, we use squared exponential kernel $c(\boldsymbol{x}, \boldsymbol{x}') = \exp(-||\boldsymbol{x} - \boldsymbol{x}'||^2/4\kappa)$ for both GP and GLGP.

4. Hyperparameters for GP or GL-GP. For positive/ non-negative continuous parameters, we put log-normal priors on them. The discrete parameter $K$ in GL-GP are pre-fixed before entering MCMC.

## B    MCMC DETAILS

Here, we provide details for MCMC iterations. The MATLAB code can be found in supplementary material, modified from Pierce (2016) and Wu (2022). For ease of sampling, we equivalently write $\boldsymbol{\eta}_j = \psi(\boldsymbol{x}_j) + \boldsymbol{\nu}_j$, with $\boldsymbol{\nu}_j \sim N_k(0, I_k)$. The full conditional distributions for parameter $\theta$ is generally noted as $\theta \mid \ldots$. Then, in each MCMC iteration:

Step 0: (for Poisson case only). Sample pseudo response $\zeta_{ij}$ by Pólya-Gamma data augmentation technique, approximating the Poisson distribution by negative binomial distribution with sufficiently large dispersion parameter $r_{ij}$.

   (a) Sample PG variable $\omega_{ij} \mid \ldots \sim \text{PG}(r_{ij} + y_{ij}, \zeta_{ij} - \log r_{ij})$, where $\text{PG}(a, b)$ denotes the Pólya-Gamma distribution abd $\zeta_{ij}$ is the sample from previous iteration.

   (b) Sample $\zeta_{ij} \mid \ldots \sim N(m_{ij}, V_{ij})$, where $V_{ij} = (\omega_{ij} + \sigma_i^{-2})^{-1}$, $m_{ij} = V_{ij}(\kappa_{ij} + \sigma_i^{-2}\mu_{ij})$ and $\kappa_{ij} = (y_{ij} - r_{ij})/2 + \omega_{ij}\log(r_{ij})$.

Step 1: Sample $\sigma_i^2$. $\sigma_i^{-2} \mid \ldots \sim \text{Gamma}\left(a_\sigma + \frac{n}{2}, b_\sigma + \frac{1}{2}\sum_{i=1}^{n}(\zeta_{ij} - \theta_i.\xi(\boldsymbol{x}_j)\boldsymbol{\eta}_j)^2\right)$, for $i = 1, \ldots, n$.

Step 2: Sample $\theta_{i\cdot}$. The full conditional for row $i$ of $\Theta$ is $\theta_{i\cdot} \mid \ldots \sim N_L\left(\Sigma_\theta \tilde{\boldsymbol{\eta}}' \sigma_i^{-2}(\zeta_{1j}, \ldots, \zeta_{nj})', \Sigma_\theta\right)$, where $\tilde{\boldsymbol{\eta}} = \{\xi(\boldsymbol{x_1})\boldsymbol{\eta_1}, \ldots, \xi(\boldsymbol{x_n})\boldsymbol{\eta_n}\}'$ and $\Sigma_\theta = \left(\sigma_i^{-2}\tilde{\boldsymbol{\eta}}'\tilde{\boldsymbol{\eta}} + \text{diag}(\phi_{i1}\tau_1, \ldots, \phi_{iL}\tau_L)\right)^{-1}$

Step 3: Sample hyper-parameters for $\Theta$. $\phi_{il} \mid \ldots \text{Gamma}(2, \frac{\gamma + \tau_l \theta_{il}^2}{2})$, $\delta_1 \mid \ldots \sim \text{Gamma}(a_1 + \frac{nL}{2}, 1 + \frac{1}{2}\sum_{l=1}^{L} \tau_l^{(-1)} \sum_{i=1}^{n} \phi_{il}\theta_{il}^2)$ and $\delta_h \mid \ldots \sim \text{Gamma}(a_2 + \frac{n(L-h+1)}{2}, 1 + \frac{1}{2}\sum_{l=1}^{L} \tau_l^{(-h)} \sum_{i=1}^{n} \phi_{il}\theta il^2)$, where $\tau_l^{(-h)} = \prod_{t=1, t\neq h}^{l} \delta_t$ for $h = 1, \ldots n$.

Step 4: Sample $\psi_m$. By rewriting $\boldsymbol{\eta}_j = \psi(\boldsymbol{x}_j) + \boldsymbol{\nu}_j$ and denoting $\Gamma_j = \Gamma(\boldsymbol{x}_j)$, we have $\boldsymbol{\zeta}_j = \Gamma_j \psi(\boldsymbol{x}_j) + \Gamma_j \boldsymbol{\nu}_j + \boldsymbol{\epsilon}_j$. Marginalizing out $\boldsymbol{\nu}_i$, $\boldsymbol{\zeta}_j \sim N(\Gamma_j \psi(\boldsymbol{x}_j), \tilde{\Sigma}_j = \Gamma_j \Gamma_j' + \Sigma_0)$. Since we put (GL-)GP prior on $\psi_l$, such that $\psi(\boldsymbol{x}_1), \ldots, \psi(\boldsymbol{x}_p) \sim N(\boldsymbol{0}, K)$, then,

$$(\psi_l(\boldsymbol{x}_1), \ldots \psi_l(\boldsymbol{x}_p))' \mid \ldots \sim N\left(\Sigma_\psi \left(\Lambda_{1l}' \tilde{\Sigma}_1^{-1} \tilde{\boldsymbol{\zeta}}_1^{(-l)}, \ldots, \Lambda_{pl}' \tilde{\Sigma}_p^{-1} \tilde{\boldsymbol{\zeta}}_p^{(-l)}\right)', \Sigma_\psi\right)$$

,where $\tilde{\boldsymbol{\zeta}}_j^{(-l)} = \boldsymbol{\zeta}_l - \sum_{r\neq l} \Gamma_{jr}\psi_r(\boldsymbol{x}_j)$ with $\Gamma_{jl}$ be $l$th column vector of $\Gamma_j$, and $\Sigma_\psi = \left(K^{-1} + \text{diag}(\Lambda_{1l}' \tilde{\Sigma}_1^{-1} \Lambda_{1l}, \ldots, \Lambda_{pl}' \tilde{\Sigma}_p^{-1} \Lambda_{pl})\right)^{-1}$

Step 5: Sample $\boldsymbol{\nu}_j$ Let $\tilde{\boldsymbol{\zeta}}_j^{-l} = \boldsymbol{\zeta}_j - \Gamma_j \psi(\boldsymbol{x}_j)$, such that $\boldsymbol{\zeta}_j = \Gamma_j \boldsymbol{\nu}_j + \boldsymbol{\epsilon}_j$, then $\boldsymbol{\nu}_j \mid \ldots \sim N((I + \xi(\boldsymbol{x}_j)'\Theta'\Sigma_0^{-1}\xi(\boldsymbol{x}_j)^{-1}\xi(\boldsymbol{x}))^{-1} \xi(\boldsymbol{x}_j)'\Theta'\Sigma_0^{-1}\tilde{\zeta}_j, (I + \xi(\boldsymbol{x}_j)'\Theta'\Sigma_0^{-1}\xi(\boldsymbol{x}_j)^{-1}\xi(\boldsymbol{x}))^{-1})$

Step 6: Sample (GL)-GP hyperparameters for $\psi$. Sample by HMC, based on Gaussian likelihood. The log-normal prior is used for positive/ non-negative parameters.

Step 7: Sample $\boldsymbol{\xi}$ Although we can sample $\boldsymbol{\xi}(\boldsymbol{x}_j)$ similar to $\psi(\boldsymbol{x}_j)$, it can be very cumbersome for data with large sample size ($L \times k \times p$). Here, we sample $\zeta_{lm}(\boldsymbol{x}_j)$ sequentially when fixing the remaining parameters in $\zeta(\boldsymbol{x}_j)$. Therefore, the problem reduced to update sequentially for regular GP regression.

Step 8: Sample (GL)- GP hyperparameters for $\boldsymbol{\xi}$. Again, we use the HMC for sampling, but only use the $\boldsymbol{\xi}$ corresponding with $\Theta$ that is large enough from 0 for hyper-parameter sampling.

## C RUNNING TIME FOR EACH SIMULATION

In simulations, the training response for both continuous and counting cases has $N = 50$ and $p = 100$, and use $q = 2$ covariates. The following two tables (Table C for continuous response and Table C for counting response) show the time consumed for each iteration, under different response types and latent dimensions. The "-fixed" means the hyperparameters for kernel function is fixed, while "-adaptive" means they are sampled in MCMC.

Table 1: **Running time for simulation with continuous response**

|  | L-GP-fixed | L-GP-adaptive | L-GLGP-fixed | L-GLGP-adaptive |
|---|---|---|---|---|
| $k = 2, L = 10$ | 0.12s | 0.20s | 0.13s | 0.21s |
| $k = 5, L = 10$ | 0.18s | 0.28s | 0.22s | 0.25s |

Table 2: **Running time for simulation with counting response**

|  | L-GP-fixed | L-GP-adaptive | L-GLGP-fixed | L-GLGP-adaptive |
|---|---|---|---|---|
| $k = 2, L = 10$ | 0.31s | 0.38s | 0.32s | 0.75s |
| $k = 5, L = 10$ | 0.37s | 0.45s | 0.37s | 0.75s |

## D SUPPLEMENTARY RESULTS FOR SIMULATIONS

In this section, we provide supplementary results for simulations in Section 3. The simulation is replicated six times. For each simulation, we observe data from 100 locations and we need to

predict response in other 1000 locations (test). The details of data generation and models can be found in Section 3. Section D.1 summarizes results for all six experiments, in terms of held-out log-likelihood for all latent factor models. The supplementary results corresponding to one set of experiment in Figure 1 are shown in Section D.2.

## D.1 HELD-OUT LOG-LIKELIHOOD

For all six independent experiments, we calculate log-likelihood for all latent models, with Gaussian response in Table D.1 and Poisson response in Table D.1, to show the robustness of our methods. The evaluations for GPWP methods (Nejatbakhsh et al., 2023) are dropped, since hyper-parameter tuning via cross-validation can be cumbersome and difficult, especially for Poisson response. For all fitted GPWP models with tuned hyper-parameters, the held-out log-likelihoods are $-6 \times 10^6$ for Gaussian response and $-10 \times 10^4$.

Table 3: **Gaussian held-out log-likelihood** For each experiment, we observe data from 100 locations and calculate the held-out likelihood for 1000 locations.

| True($\times 10^4$) | L-GP($\times 10^4$) | | L-GLGP-fixed($\times 10^4$) | | L-GLGP-adptive($\times 10^4$) | |
|---|---|---|---|---|---|---|
| | k=2 | k=5 | k=2 | k=5 | k=2 | k=5 |
| 8.57 | 8.14 | 8.15 | 8.17 | 8.17 | 8.24 | 8.22 |
| 8.55 | 8.13 | 8.07 | 8.14 | 8.08 | 8.17 | 8.14 |
| 8.56 | 8.15 | 8.16 | 8.21 | 8.16 | 8.22 | 8.20 |
| 8.56 | 8.18 | 8.22 | 8.18 | 8.24 | 8.28 | 8.25 |
| 8.57 | 8.11 | 8.08 | 8.17 | 8.12 | 8.25 | 8.13 |
| 8.56 | 8.18 | 8.14 | 8.20 | 8.15 | 8.23 | 8.20 |

Table 4: **Poisson held-out log-likelihood** For each experiment, we observe data from 100 locations and calculate the held-out likelihood for 1000 locations.

| True($\times 10^4$) | L-GP($\times 10^4$) | | L-GLGP-fixed($\times 10^4$) | | L-GLGP-adptive($\times 10^4$) | |
|---|---|---|---|---|---|---|
| | k=2 | k=5 | k=2 | k=5 | k=2 | k=5 |
| -4.49 | -4.64 | -4.79 | -4.63 | -4.67 | -4.61 | -4.62 |
| -4.49 | -4.63 | -4.65 | -4.62 | -4.62 | -4.62 | -4.61 |
| -4.47 | -4.69 | -4.75 | -4.68 | -4.72 | -4.67 | -4.69 |
| -4.49 | -4.62 | -4.65 | -4.60 | -4.64 | -4.60 | -4.61 |
| -4.49 | -4.63 | -4.67 | -4.61 | -4.62 | -4.61 | -4.61 |
| -4.47 | -4.77 | -4.75 | -4.74 | -4.71 | -4.69 | -4.67 |

## D.2 FITTED MEAN AND COVARIANCE

The results shown here correspond to the experiment in Figure 1. The Figure A1 provides fitted mean and covariance for three models (L-GP and L-GLGP-fixed/adaptive) in the PC2 and PC3 for Gaussian (Fig. A1A) and Poisson response (Fig. A1B), when $k = 2$ (ground truth) and $L = 10$. To study the sensitivity of misspecified latent dimension, we refit models with $k = 5$ and $L = 10$, and plot the fitted mean and covariance to PC space. The Gaussian response is relative roust to misspecified $k$ (Fig. A2), while the effect on Poisson response is relatively significant (Fig. A3).

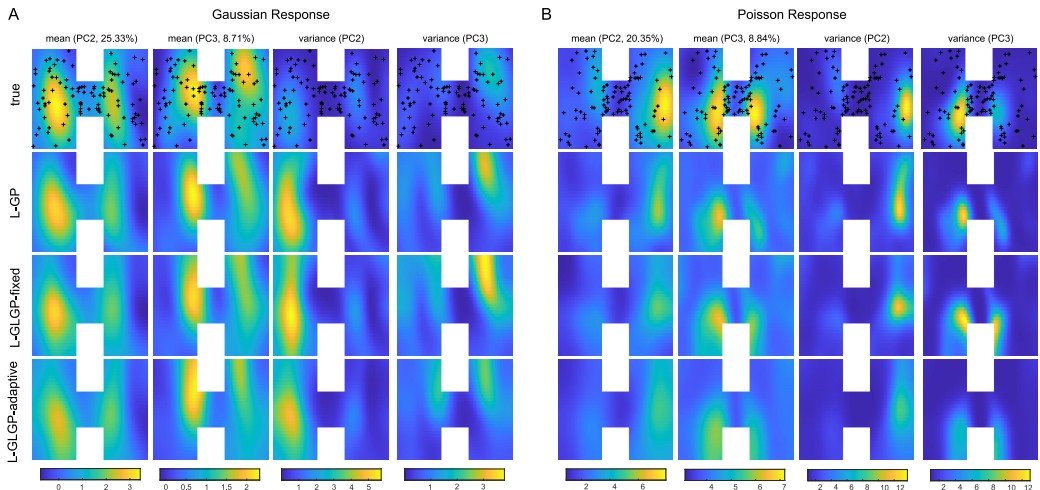

Figure A1: **Supplementary results for L = 10 and k = 2.** The true and fitted mean and covariance in second and thrid PC space, for L-GP, L-GLGP-fixed and L-GLGP-adaptive models, using $L = 10$ and $k = 2$. The results of Gaussian response in (A) and Poisson response in (B). The observed locations are overlaid, and the variances explained by PCs are shown alongside.

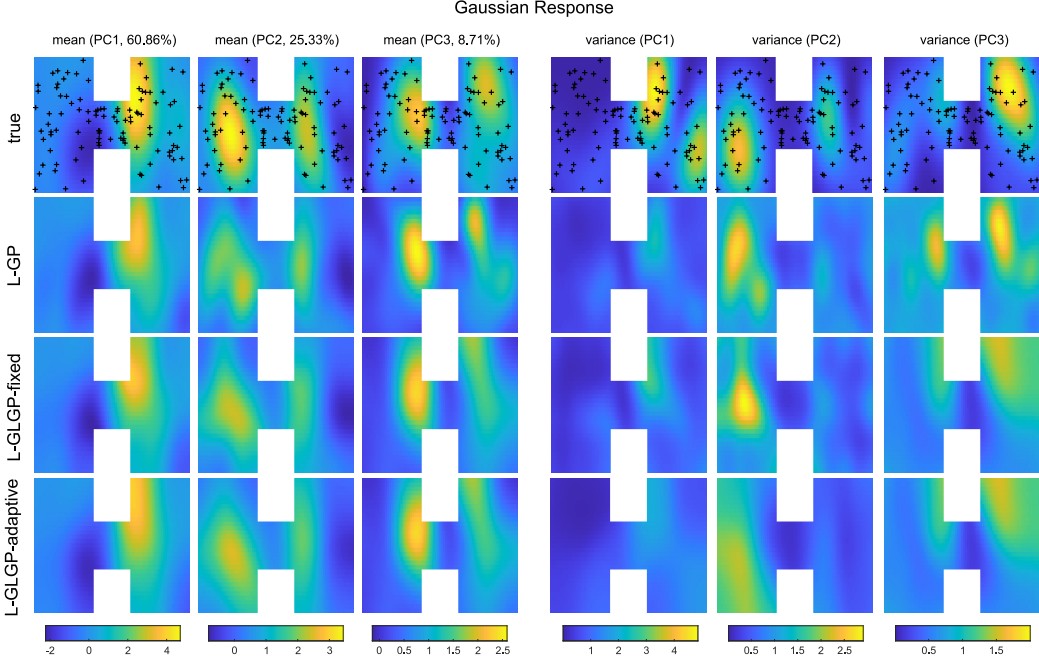

Figure A2: **Results of Gaussian response for L = 10 and k = 5.** The true and fitted mean and covariance of Gaussian response in the first three PCs space, for L-GP, L-GLGP-fixed and L-GLGP-adaptive models, using $L = 10$ and $k = 5$. The observed locations are overlaid, and the variances explained by PCs are shown alongside.

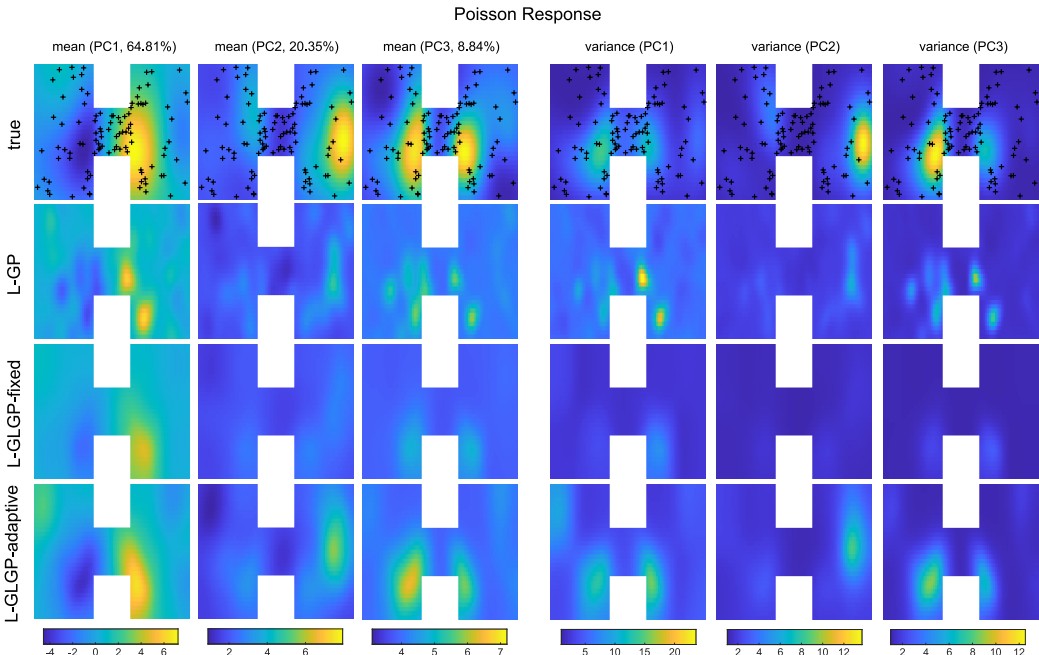

Figure A3: **Results of Poisson response for L = 10 and k = 5.** The true and fitted mean and covariance of Poisson response in the first three PCs space, for L-GP, L-GLGP-fixed and L-GLGP-adaptive models, using $L = 10$ and $k = 5$. The observed locations are overlaid, and the variances explained by PCs are shown alongside.