# OpenReview forum: "Nonparametric Covariance Regression for Massive Neural Data on Restricted Covariates via Graph"
_ICLR.cc/2025/Conference — Submitted to ICLR 2025_

### Official Review · Reviewer_DLGM · 2024-10-31

**Soundness:** 3
**Presentation:** 2
**Contribution:** 2
**Rating:** 5
**Confidence:** 3

**Summary:**

The authors present a non-parametric covariance regression method that allows modeling non euclidean via the  Graph Laplacian Gaussian process proposed by Dunson et al. 2022. They perform numerical experiments on simulated and neuron data.

**Strengths:**

The paper is sound, correctly written, and well-motivated. Overall, the work reads solid, and the method proposed seems to work relatively well.

**Weaknesses:**

The paper essentially relies on the work of Dunson et al. and the presented work seems to be closer to an adaptation of this work than a novel prior /method in the context of covariance regression. While this is interesting in principle, the method contribution seems a bit limited to a slight variation of the original work of Dunson and colleagues. Similarly, the authors do not tackle any particularly novel complex sampling scheme for their MCMC part and rely on existing work that they blend with the GL-GP prior to Dunson and colleagues.

 Further, the simulations are somehow limited (perhaps due to the heaviness of the computation for inference).

Figure 6 seems to advocate the most for the method, but Figures 1, 4, and 5 do not strike me as proper improvements.
Lastly, it would have been nice to compare with non-GL-GP methods (perhaps a standard vanilla GP approach).

**Questions:**

Have run similar experiment using some of the competing methods?
Could you clarify the data size (covariate and neurons data).

---

> ### Author Response · Authors · 2024-11-18
>
> Thanks a lot to the reviewer for their constructive comments. Here, we clarified some specific points…
>
> 1. > The paper essentially relies on the work of...
>
> See general comments 1.
>
> 2. > Further, the simulations are somehow limited ...
>
> For number of neurons ($N$, 'massive neurons'), the method doesn't have scalability issue, since we assume these neurons share the same set of low dimensional latent factors. The multiplicative shrinkage prior allows us adaptively choose $L$.
>
> For number of experiement conditions (recording length for time series), current implementation can be cumbersome, since we need large matrix inversion for GP. This can be a future direction, for example do approximated inference (e.g. variational inference) or use some trick to reduce computation cost for GP matrix inversion (e.g. sparse GP). For more detailed discussion, see general comments 2.
>
> 3. > Figure 6 seems to advocate the most for the method...
>
> The counting data is more challenging than the continuous data, and this may explain why Figure 6 shows more significant results. Even for the remaining simulations, the improvement can be shown in held-out likelihood. See more detailed discussion in general comments 3.
>
>
> 4. > Lastly, it would have been nice to compare with non-GL-GP methods...
>
> Currently, there are not a lot of methods for mean-covariance regression in computational neuroscience community, and we compare our methods to GPWP and dCMP at this point. May think about comparisons to other methods in broader statistical community whenever possible.
>
>
> 6. > Could you clarify the data size (covariate and neurons data)?
>
> The details of data size can be found in our updated manuscript and general comments 2.

---

### Official Review · Reviewer_1hvp · 2024-11-03

**Soundness:** 3
**Presentation:** 2
**Contribution:** 2
**Rating:** 5
**Confidence:** 4

**Summary:**

The authors present a framework for modeling neural data and covariates through including graph properties of covariates to account for restricted subspaces. They demonstrate their method in simulation and in two experimental datasets.

**Strengths:**

- The inclusion of accounting for restricted space / inputs is smart and very relevant for neuroscience applications in particular.
- The model components and math are well explained individually, though the thread of motivation throughout the methods section could be strengthened to connect back to the overall picture.
- Simulations, training, selection of parameters, and results on experimental datasets are all very clear with a high level of detail (needed for any future work or replication, which is excellent).

**Weaknesses:**

- The results overall are underwhelming. In the simulated data, all model fits look extremely similar. How much of the observed variation would be needed in a neuroscience context? In the fits to experimental data, the significance of the results is not explained. The PC plots are results, but what do they tell us? What new findings did we learn from this method of modeling the neural data?
- Computational costs are discussed but not shown empirically.
- Minor notes: typo (extra ')') in line 66
- References to e.g. Figure 4 (which is really in the supplement) are confusing.
- Grammar throughout could use some editing for correctness and clarity (e.g., the sentence spanning lines 82-84 doesn't make sense to me).
- The provided code is complex (many files/functions) and lacks documentation, and is in Matlab, which lessens its overall impact in the ML field (and likely the efficiency and speed of computation for MCMC).

**Questions:**

- The code implementation appeared custom. Why not use established packages where appropriate (e.g. MCMC)?
- "we observe that the inference can be sensitive to hyper-parameters ({, K, t}) for GL-GP" (line 222) Is there evidence to show here?
- How does this scale to higher numbers of latent factors and increased dimensionality?  The simpler experiments showed "L = 10 is large enough" (line 262), so perhaps it is not a concern for low-dimensional tasks, but I would be curious to know the authors' thoughts on scalability (to tasks involving more freely moving behaviors, for example).
- The observed data for the simulations in Figure 1 were fairly widely spread out throughout the space. How well would this method work for the cases where test or new data extended outside the range of previous observations?

---

> ### Author Response · Authors · 2024-11-18
>
> Thanks a lot to the reviewer for their constructive comments. Here, we clarified some specific points…
>
> 1. > The results overall are underwhelming...
>
> See detailed discussion in general comments 3. The first 2-3 PCs capture the majority of variability, and the interpretation in the context of neuroscience is provided in the manuscript.
>
> 2. > Computational costs are discussed but not shown empirically...
>
> See general comments 2.
>
> 3. > The provided code is complex (many files/functions) and lacks documentation...
>
> Thanks for suggestions, and it's worthwhile to document the code more carefully to make things more readable. Implementing the model into Python is also a good suggestion, but this may be out of scope of this paper.
>
> 4. > The code implementation appeared custom...
>
> The major reason is that we want our sampling be tractable, rather than general sampling method like HMC. For example, the established package may not provide the P\'olya-Gamma (PG) augmentation at this point.
>
> 5. > we observe that the inference can be sensitive to hyper-parameters...
>
> Please refer to simulations, especially figure 1B and 1E, and corresponding texts.
>
> 6. > How does this scale to higher numbers of latent factors and increased dimensionality...
>
> For number of neurons ($N$, 'massive neurons'), the method doesn't have scalability issue, since we assume these neurons share the same set of low dimensional latent factors. The multiplicative shrinkage prior allows us adaptively choose $L$.
>
> For number of experiement conditions (recording length for time series, i.e., $p$), current implementation can be cumbersome when this is large, since we need large matrix inversion for GP. This can be a future direction, for example do approximated inference (e.g. variational inference) or use some tricks to reduce computation cost for GP matrix inversion.
>
>
> 7. > The observed data for the simulations in Figure 1 were fairly widely spread out throughout the space...
>
> The performance may depend on GP kernel we choose. But all kernel functions can be tailored to GLGP, but GL further make use of the geometric information (which presumed to be better).

---

### Official Review · Reviewer_wnZ7 · 2024-11-04

**Soundness:** 3
**Presentation:** 2
**Contribution:** 2
**Rating:** 6
**Confidence:** 3

**Summary:**

This work proposes a nonparametric mean-covariance regression model that leverages latent factor analysis and Gaussian Processes (GPs) to jointly model the mean and covariance of the high-dimensional neural data. Despite the a bit hard to follow writing flow, a notable claimed contribution is the integration of graph-based Gaussian processes to manage covariates in resricted subspaces with the MCMC algorithm. The model is validated through simulation dataset and applied to real-world LFP and HC neural datasets.

**Strengths:**

1. The introduction of graph-based Gaussian processes to handle restricted covariates adds a novel dimension to mean-covariance modeling, which I believe is helpful for neural data. Because this approach can be effective while maintaining the interpretability of the method.
2. The authors employ an MCMC algorithm with data augmentation techniques to handle intractable count models effectively, which can be more computationally effficient.
3. The authors argued and claimed that this proposed framework can be extended beyond neuroscience resaerch to other domains requiring mean-covariance modeling under similar constraints.

**Weaknesses:**

1. The paper is a bit hard to follow in writing, the scientific question or mathematical task you hope to solve is not very clear.
2. There has the assumption of independent GPs for each factor dimension, although computationally efficient, might overlook some inherenet underlying dependencies between factors that could be relevant in most neuroscience applications.

**Questions:**

1. What's the potential broader applications of this framework beyond neuroscience?
2. My other concerns please relate to the Weakness section.

---

> ### Author Response · Authors · 2024-11-18
>
> Thanks a lot to the reviewer for their constructive comments. Here, we clarified some specific points…
>
> 1. > The paper is a bit hard to follow in writing...
>
> As shown in the title, the goal for this paper is to provide a framework to do mean-covariance regression for high dimensional neural data (in terms of number of neurons) in latent space, where constraint of input is handled by graph information and correlation/ smoothness over different experiement conditions is handled via GP.
>
>
> 2. > There has the assumption of independent GPs for each factor dimension...
>
> The main purpose of this paper is to build a general framework, and most factor analysis based on GP (e.g. GPFA) use the independent assumption. Surely, we can improve the framework using dependent GPs (DGP). Recently, DGP factor analysis has been used in gene expression data ([1]), where they found DGP can improve estimation, especially the factors are truly correlated and individual signals are weak. However, for methods with prespecified latent dimension, the correlated latent factor may suggest redundant dimension. If we infer number of latent factor by BNP or some shrinkage prior, the independence assumption should be appropriate to some degree. But the reviewer is correct, for some problems, we may know/ need to prespecify latent dimension based on field knowledge, and using DGP allows us to infer correlation structure of these factors.
>
> 3. > What's the potential broader applications of this framework beyond neuroscience?
>
> The framework is built for high dimensional mean-covariance regression with restricted input, for both continuous and counting response. Therefore, it can be applied to any data with the same structure, especially for spatio(temporal) data, which is usually high dimensional and has spatial restriction.

---

### Official Review · Reviewer_4Jkz · 2024-11-08

**Soundness:** 3
**Presentation:** 3
**Contribution:** 2
**Rating:** 5
**Confidence:** 3

**Summary:**

The authors consider modeling neuroscience data (such as LFP and neural spike count data) in settings where experimental conditions can vary in complicated ways (that may not be well-modeled by assuming conditions lie in a Euclidean space, for instance).  They propose a non-parametric mean-covariance regression model.   They impose smoothness along both experimentation conditions and latent factors. To handle restricted domains they incorporate Graph Laplacian (GL) kernels.  For count data they handle non-conjugate count likelihood using a data augmentation technique.  In the inference step, they propose an MCMC algorithm with a data augmentation technique and derive full conditional to get posterior samples. They consider both continuous (LFP) and count observation real world neuroscience datasets.

**Strengths:**

## Strengths
- Analyzing neuroscience data (esp. count also LFP) is challenging but important.
- The problem of handling restricted spaces arising from experimental conditions is well-motivated.
- The use of a graph-Laplacian GP to allow for more flexibility than assuming an unknown manifold is interesting.
- The presentation overall was good.
- Experiments are included from two real-world neuroscience experiments, one on LFP data and one for count data.

**Weaknesses:**

## Weaknesses
### Major
- My major concern regards technical novelty.  From my reading, the results seem to largely follow from integrating two key prior works, Fox and Dunson 2015 which proposed  predictor-dependent factor loadings modeled using GPs and Dunson et al 2021 which studied Graph Laplacian based GP regression for restricted domains.  Perhaps the authors could elaborate on any significant technical challenges that were overcome.
- The title includes “MASSIVE NEURAL DATA” and “massive” is mentioned several times but not described in particular with respect to computational or sample complexities
    – can you elaborate what ranges of which data dimensions qualify as “massive”?
    - The computational complexity of the proposed method is not discussed, especially important in settings with massive data; how does the complexity vary with different data dimensions?
    -  Relatedly, empirical run-times for your experiments and the GPWP baseline are not reported
    - sample complexity is not analyzed theoretically or experimentally
    - The HC experiment involved 36 neurons in one recording session  with 200 ms bins.  The LFP data set I am not sure of the dimensions (eg LFPs from 13 areas – how many LFPs recorded per area?), but from the description did not appear to be ``massive.’’
    - (minor) in the discussion, a point is made regarding computational complexity in lines 467-470 “in applications to massive data” which sounds like saying in settings with even larger data dimensions than is considered here


### Minor
- For the GPWP baseline (Nejatbakhsh et al., 2023), the authors mention the used "single trial", but did not include further discussion on the choice.
- The GPWP reported value in the simulation experiment for the Gaussian case differs dramatically – any thoughts why that is?
- The L-GP baseline’s scores in most experiments (to me) seemed pretty close to the proposed method.


### Very minor (Typos/etc.)
- line 71 ‘massive neurons.’
- 081 "(p >> N)" although standard notations, p and N are have not been introduced before this line
- a few commas start a new line after equation blocks (eg line 148)
- line 196 ‘identifiablility.’

**Questions:**

(I have some doubts related to technical novelty, computational complexity, sample complexity, and baselines mentioned above -- clarification on those points would be appreciated)

---

> ### Author Response · Authors · 2024-11-18
>
> Thanks a lot to the reviewer for their constructive comments, and have updated our manuscript accordingly. Here, we clarified some specific points…
>
> 1. > My major concern regards technical novelty...
>
> See general comments 1.
>
> 2. > The title includes “MASSIVE NEURAL DATA” and “massive”...
>
> The "massive" refers to number of neurons. **This motivates us to do everything in latent space** (to reduce dimension), and hence we can handle large amount of neurons simultaneously.
>
> 3. > The computational complexity...
>
> See general comments 2 for more detailed discussion. The reviewer correctly notes that these applications are not "massive" in terms of neuron count, but these relatively small dataset (considering computing source and time) is used here to illustrate usage of our method(s) conveninently. Current application example fits the illustrative purpose well, but full neural data analysis can make the application part more solid, and we may consider it whenever possible later.
>
> Moreover, the number of neurons does not significantly impact computation, because we use a latent factor model for dimensionality reduction. At present, the main bottleneck for computation time is the number of experimental conditions (recording length for time series), as we require large matrix inversions for the GP.
>
> For GPWP, the fitting with fixed parameters is fast. But selecting tuning parameters by cross-validation can be painful.
>
> 4. > GPWP...
>
> The GPWP is designed for repeated trials. Since GPWP doesn't do things in latent space, the correlation among neurons is ignored, and hence it leads to worse estimation.
>
> 5. > The L-GP baseline’s scores in most experiments (to me) seemed pretty close to the proposed method...
>
> The L-GP is not necessarily a baseline (the baseline should be non latent models, such as GPWP), although it has been developed for continuous response. The application to counting response is new (with PG augementation). For more detailed discussion, see general comments 3.

---

### Author Response · Authors · 2024-11-18
**General Comments**

Thanks all the reviewers for their constructive comments, and we have updated the manuscript accordingly. Here, we try to clarify the following three points raised by most reviewers.

## 1. Novelty

Overall, the primary purpose of this paper is to introduce a new method and perspective for addressing neuroscience problems for large number of neurons/ subjects. Previously, researchers mainly focused on dimension reduction (e.g., LDS and GPFA), which can at most parse to mean regression problem. Meanwhile, many mean-covariance regression methods do not account for dimension reduction. In addition to merging these two perspectives, we incorporate the geometric information of restricted inputs to enhance performance.

Besides, another significant contribution is providing a tractable inference method for count data (e.g., neural spikes) via P\'olya-Gamma (PG) augmentation. Furthermore, here are some additional points we'd like to highlight. 1) Combining building blocks is non-trivial for neural data analysis, as brute-force inference becomes infeasible for lengthy recordings. Here, we sequentially sample $\xi(x)$. 2) We also observed that while all models show relative robustness for Gaussian responses, the fitting results for count responses are sensitive to the latent dimension $k$ in L-GP and L-GLGP-fixed, while L-GLGP-adaptive model is not.This empirically suggests the need to consider all effects in count data analysis.

## 2. Computational Complexity

The computational xomplexity was added to the updated manuscript. Basically, all the following results are from a laptop with an Intel(R) Core(TM) i7-8665U CPU @ 1.90GHz 2.11 GHz.

In simulations, the training response for both continuous and counting cases has $N = 50$ and $p = 100$, and use $q=2$ covariates. The following two tables show the time consumed for each iteration, under different response types and latent dimensions. The "-fixed" means the hyperparameters for kernel function is fixed, while "-adaptive" means they are sampled in MCMC.

For continuous response:

|  | L-GP-fixed | L-GP-adaptive | L-GLGP-fixed| L-GLGP-adaptive|
|----------|----------|----------|----------|----------|
| $k=2, L = 10$ | 0.12s | 0.20s | 0.13s | 0.21s |
| $k=5, L = 10$ | 0.18s | 0.28s | 0.22s | 0.25s |


For counting response:

|  | L-GP-fixed | L-GP-adaptive | L-GLGP-fixed| L-GLGP-adaptive|
|----------|----------|----------|----------|----------|
| $k=2, L = 10$ | 0.31s | 0.38s | 0.32s | 0.75s |
| $k=5, L = 10$ | 0.37s | 0.45s | 0.37s | 0.75s |

In terms of application to neural data: 1) For LFP with a continuous response, the training response matrix is $14 \times 700$, and the covariate dimension is 3. Each iteration takes ~3.5 seconds. 2) For HC with a counting response, the training response matrix is $36 \times 48$, and the covariate dimension is 2. Each iteration takes ~3.3 seconds.

The number of neurons does not significantly impact computation (as in simulations), because we use a latent factor model for dimensionality reduction. At present, the main bottleneck for computation time is the number of experimental conditions (recording length for time series), as we require large matrix inversions for the GP. For application to data with long recording/ large number of experiment conditions we suggest to do approximated inference (e.g. variational inference) or use some trick to reduce computation cost for GP matrix inversion (e.g. sparse GP or [1]), as mentioned in the discussion.

## 3. Performance among different methods

All latent models with smoothness,by (GL)GP, over different conditions (L-models) are somewhat similar in terms of overall metrics such as held-out log-likelihood (although in LFP application, L-GLGP-adaptive improves more significantly).

However, 1) these L-models are significantly better than non latent models, such as GPWP and dCMP; 2) a factor model without smoothness (e.g. by GP) is consistent only when $p/N\to 0$ ([2]). These suggest that considering dimension reudction and smoothness (to some degree, reduce effective $p$) over experiement conditions/ time can handle most cases, and the overall performance of GP and GLGP are similar (this also applies to the original GLGP paper, Dunson et al 2022).

**However**, some local performance (estimation at specific conditions/ time points/..., e.g. especially in figure A3) can be quite different for different models, and for problems requiring better local inference, considering geoemtric information is necessary. To make things more visually significant, we can design a more complicated geometric constraint.

[1] Radial neighbours for provably accurate scalable approximations of Gaussian processes, Zhu et al., Biometrika, 2024

[2] On Consistency and Sparsity for Principal Components Analysis in High Dimensions, Johnstone I.M. & Lu, A.Y., Journal of the American Statistical Association, 2009

---

### Meta-Review · Area_Chair_fdDY · 2024-12-18

**Metareview:**

This paper proposes a non-parametric mean-covariance regression model for neural data analysis, integrating Graph Laplacian Gaussian Processes (GL-GPs) to address restricted experimental subspaces. Validated on simulated datasets and real-world neural data (LFP and spike counts), the method employs MCMC-based inference, showcasing versatility across data types.

Despite this, the paper suffers from poor clarity in notation, insufficient justification for key modeling choices, and overstated claims regarding scalability for "massive neural data." The further factorization of latent components is not well-motivated, and the kernel definitions for Gaussian processes lack transparency, leaving computational complexity underexplored. Additionally, the scalability of the inference process is questionable, given the challenges of GP-based methods when handling large datasets or time bins. These factors, coupled with unclear writing and underwhelming results, diminish the paper’s overall impact. I recommend rejection due to the limited technical novelty and these critical issues.

**Additional Comments On Reviewer Discussion:**

The notations in the paper are poorly defined. For instance, when \mu is first introduced, its meaning is not explained, and this lack of clarity extends to nearly all notations. What do these terms represent in the context of neural data analysis? The factorization of \mu into \Lambda and \psi is already a low-dimensional decomposition, but the further factorization of \Lambda into \Theta and \xi is not well justified. If such a step is necessary, why not simply set k to be small enough, since the primary goal of latent dimension reduction is to manage complexity arising from "massive neurons"? The claim of handling "massive neurons" is overstated. If the justification relies on the loading matrix being time-dependent, this needs to be explicitly argued, rather than merely referencing a Wishart process from another paper. It is unreasonable to expect readers to consult external sources to understand fundamental modeling choices within this paper.

The kernel definitions for the two Gaussian processes (GPs) are also unclear. What are the input spaces of these kernel functions, and how large are the resulting Gram matrices? This lack of clarity also leaves the mathematical analysis of computational complexity underexplored. In the rebuttal, the authors provide a table of practical running times, but these are derived from simulations, and the number of time points used remains unspecified. This omission is significant, as the number of time points is often the primary bottleneck preventing GPs from scaling effectively.

Moreover, during the inference step, challenges arise when working with GPs: sequentially sampling \xi(x), tuning sensitive hyperparameters, and introducing Polya-Gamma augmentation with auxiliary variables. These complexities, especially when the number of time bins becomes large (easily reaching hundreds in real neural datasets), raise doubts about the scalability of the proposed approach. While some issues raised by reviewers have been partially addressed, others—like these critical concerns—remain unaddressed. These unresolved factors significantly diminish the likelihood of this paper being accepted.

---

### Decision · Program_Chairs · 2025-01-22

Reject